# Punicalagin Ameliorates Lupus Nephritis via Inhibition of PAR2

**DOI:** 10.3390/ijms21144975

**Published:** 2020-07-14

**Authors:** Yohan Seo, Chin Hee Mun, So-Hyeon Park, Dongkyu Jeon, Su Jeong Kim, Taejun Yoon, Eunhee Ko, Sungwoo Jo, Yong-Beom Park, Wan Namkung, Sang-Won Lee

**Affiliations:** 1College of Pharmacy, Yonsei Institute of Pharmaceutical Sciences, Yonsei University, Incheon 21983, Korea; ddukdae12@gmail.com (Y.S.); armisael1990@gmail.com (D.J.); dsdyu2005@naver.com (S.J.); 2New Drug Development Center, Daegu-Gyeongbuk Medical Innovation Foundation, Daegu 41061, Korea; 3Division of Rheumatology, Department of Internal Medicine, Yonsei University College of Medicine, Seoul 03722, Korea; CHINHEEMUN@yuhs.ac (C.H.M.); sj1118@yuhs.ac (S.J.K.); tjy0627@yuhs.ac (T.Y.); eunny89@yuhs.ac (E.K.); yongbpark@yuhs.ac (Y.-B.P.); 4Graduate Program of Industrial Pharmaceutical Science, Yonsei University, Incheon 21983, Korea; sohyeon0605@hotmail.com; 5BK21 Plus Project, Department of Medical Sciences, Yonsei University College of Medicine, Seoul 03722, Korea; 6Institute for Immunology and Immunological Diseases, Yonsei University College of Medicine, Seoul 03772, Korea; 7Interdisciplinary Program of Integrated OMICS for Biomedical Science Graduate School, Yonsei University, Seoul 03772, Korea

**Keywords:** punicalagin, PAR2, podocyte, systemic lupus erythematosus, lupus nephritis, NZB/W F1 mice

## Abstract

Lupus nephritis (LN) is the most frequent phenotype in patients with systemic lupus erythematosus (SLE) and has a high rate of progression to end-stage renal disease, in spite of intensive treatment and maintenance therapies. Recent evidence suggests that protease-activated receptor-2 (PAR2) is a therapeutic target for glomerulonephritis. In this study, we performed a cell-based high-throughput screening and identified a novel potent PAR2 antagonist, punicalagin (PCG, a major polyphenol enriched in pomegranate), and evaluated the effects of PCG on LN. The effect of PCG on PAR2 inhibition was observed in the human podocyte cell line and its effect on LN was evaluated in NZB/W F1 mice. In the human podocyte cell line, PCG potently inhibited PAR2 (IC_50_ = 1.5 ± 0.03 µM) and significantly reduced the PAR2-mediated activation of ERK1/2 and NF-κB signaling pathway. In addition, PCG significantly decreased PAR2-induced increases in ICAM-1 and VCAM-1 as well as in IL-8, IFN-γ, and TNF-α expression. Notably, the intraperitoneal administration of PCG significantly alleviated kidney injury and splenomegaly and reduced proteinuria and renal ICAM-1 and VCAM-1 expression in NZB/W F1 mice. Our results suggest that PCG has beneficial effects on LN via inhibition of PAR2, and PCG is a potential therapeutic agent for LN.

## 1. Introduction

Systemic lupus erythematosus (SLE) is an autoimmune disease in which the body produces antibodies against itself, resulting in major organ damage [1]. Lupus nephritis (LN) is one of the most serious complications of SLE and is mainly induced by immune deposits in kidney glomeruli. Even with recommended therapeutic strategies, 10–20% patients with LN progress to end-stage renal disease (ESRD) within 5–10 years after diagnosis [2,3]. Several biological agents that regulate autoreactive B and T cells have been developed, but their efficacies have not yet been proven [2,4,5].

Protease-activated receptor-2 (PAR2) is a G protein-coupled receptor (GPCR) that is mainly activated by extracellular serine proteases [6,7]. PAR2 is widely expressed in human tissues, particularly in cells in the kidney, such as podocytes, mesangial cells, tubular epithelial cells, and infiltrating immune cells [8,9,10,11]. Activation of PAR2 by serine proteases in the kidney initiates and accelerates injury and collagen synthesis [12]. Furthermore, PAR2 enhances the production of pro-inflammatory cytokines, by activating extracellular signal-regulated kinase (ERK)/mitogen-activated protein kinase (MAPK) pathways and nuclear factor kappa B (NF-κB). This enhanced cytokine production ultimately leads to an increase in proliferation and inflammation in kidneys [13,14,15]. In vivo studies using PAR2-deficient mice have shown a decrease in inflammatory responses in diabetic nephropathy and glomerulonephritis [16]. Similarly, a complicated network of autoreactive T and B cells aggravate LN via cytoplasmic kinases and pro-inflammatory cytokines [4,17,18]. Given that the pathogenic mechanisms underlying LN are closely related to PAR2 signaling, PAR2 could be a critical therapeutic target. Previously identified small-molecule antagonists of PAR2 exhibited insufficient potency and specificity [19], emphasizing the need for a novel PAR2 antagonist. A potent and selective PAR2 antagonist would improve our understanding of the pathophysiological roles of PAR2 in inflammatory diseases, including LN. 

In this study, we identified a novel bioactive antagonist of PAR2, punicalagin (PCG), by a cell-based high-throughput screening approach. We characterized the effects of PCG in the human podocyte cell line and in lupus-prone NZB/W F1 mice. 

## 2. Results

### 2.1. Identification of a Novel PAR2 Antagonist, Punicalagin

To identify novel PAR2 antagonists, we performed a cell-based high-throughput screening with 1279 natural products in HaCaT cells expressing PAR2. PAR2 antagonists strongly blocked intracellular calcium elevation by trypsin-induced activation of PAR2 in HaCaT cells (Appendix A). Interestingly, punicalagin and punicalin, the major polyphenols in pomegranate, were identified as PAR2 antagonists (Figure 1A and Appendix A) [20]. In HaCaT cells, PCG (IC_50_ = 1.90 ± 0.07 µM) inhibited PAR2 activity more potently than punicalin (IC_50_ = 19.1 ± 0.04 μM) (Appendix A).

### 2.2. Specific Inhibition of PAR2 by PCG in a Human Podocyte Cell Line 

In the human podocyte cell line, PCG potently inhibited PAR2-activating peptide (PAR2-AP) and trypsin-induced PAR2 activation with IC_50_ values of 1.5 ± 0.03 and 2.4 ± 0.04 μM, respectively (Appendix A and Figure 1B). To investigate the selectivity of PCG on PAR2, we observed the effect of PCG on PAR1 activity and found that PCG is up to 30-fold more selective for PAR2 than PAR1 in human podocytes (Appendix A and Figure 1B).

### 2.3. Inhibition of the Intracellular Signaling Pathway of PAR2 by PCG 

PAR2 activation stimulates the ERK1/2 and NF-κB signaling pathways in various cell types [21]. To investigate whether the phosphorylation of ERK1/2 by PAR2 activation is inhibited by PCG, the human podocyte cell line was pre-treated with PCG and then PAR2 was activated with PAR2-AP. PCG significantly inhibited phosphorylation of ERK1/2 and NF-κB p65 protein by PAR2 activation (Figure 1C and Appendix A). In both human bronchial fibroblasts and human keratinocytes, PAR2 activation upregulates VCAM-1 and ICAM-1 via NF-κB activation [22,23]. We further investigated the effect of PAR2 activation and PCG on the expression of VCAM-1 and ICAM-1 in the human podocyte cell line. PAR2 activation significantly upregulated VCAM-1 and ICAM-1, and the upregulation was significantly inhibited by PCG (Figure 1D and Appendix A).

### 2.4. Suppression of the PAR2-Induced Upregulation of Pro-Inflammatory Cytokines by PCG

PAR2 activation induces the production of pro-inflammatory cytokines, such as IL-6, IFN-γ, TNF-α and IL-8, in various cell types [22,23]. In a pilot study on the effect of PCG on cytokine production by PAR2 activation, we observed that IL-6 production peaked at 12 h after treatment with PAR2-AP in human podocyte cell line (Appendix A). Through further analysis, we found that PCG significantly inhibited PAR2 stimulation-induced increases in the production of IL-6, IFN-γ, TNF-α and IL-8 at 12 h after PAR2-AP application (Figure 1E–H).

### 2.5. PCG Improves Survival and Proteinuria in NZB/W F1 Mice 

To examine whether PCG has a protective effect against LN, we used lupus-prone female NZB/W F1 mice, which develop a spontaneous severe autoimmune disease resembling human SLE and produce high titers of antinuclear antibodies associated with the development immune complex-mediated glomerulonephritis [24]. Intraperitoneal administration of PCG was initiated when the female NZB/W F1 mice were 23 weeks of age and continued for 7 weeks (Figure 2A). Two mice in the vehicle-treated group died of aggravation caused by LN at 28 weeks of age, but all methylprednisolone (MPL)-treated and PCG-treated mice survived till 30 weeks of age (Figure 2B). At 23 weeks of age, the proteinuria score was around 1.5+ for most lupus-prone mice. The proteinuria score gradually increased to 3.0+ at 30 weeks of age in vehicle-treated mice (Figure 2C). Mice treated with both 1 and 3 mg/kg PCG (but not 0.3 mg/kg) significantly reduced proteinuria compared to that in vehicle-treated mice. The proteinuria-reducing effect of 1 and 3 mg/kg PCG was comparable to that of MPL (Figure 2C). PCG-treated mice exhibited preserved renal function based on serum creatinine; however, MPL-treated mice showed no preservation of renal function, despite a lower serum creatinine level than that of vehicle-treated mice (Figure 2D). 

### 2.6. PCG Reduces Glomerular and Tubular Damage and Immune Deposits in Renal Glomeruli 

At 30 weeks of age, NZB/W F1 mice exhibited remarkable glomerular damage, including severe glomerular expansion, hypercellularity, inflammatory immune cell infiltration, and focal crescents along with tubular damage (Figure 3A) [25]. Interestingly, the administration of MPL and 3, 1, and 0.3 mg/kg PCG substantially reduced glomerular damage (74.9%, 68.8%, 68.8%, and 56.1%, respectively) and tubular damage (79.0%, 80.5%, 76.6%, and 70.7%, respectively) compared to those in vehicle-treated mice. However, there were no significant differences in vascular damage among mouse groups (Figure 3B). 

### 2.7. PCG Reduces Immune Deposits in Renal Glomeruli

Immunofluorescence analysis showed heavy accumulation of IgG (red) and C3 (green) in the mesangium and capillary loops within glomeruli of vehicle-treated mice (Figure 3C). Colocalization of IgG and C3 as immune complexes (merge) was clearly apparent in vehicle-treated mice. However, PCG significantly decreased the fluorescence intensities of both IgG and C3, indicators of the extent of immune-complex formation within glomeruli (Figure 3D). 

### 2.8. PCG Rebalances Serum Inflammatory Cytokines

The administration of MPL and 3, 1, and 0.3 mg/kg PCG significantly reduced serum levels of IFN-γ compared to their levels in vehicle-treated NZB/W F1 mice (Figure 4A). Serum levels of IL-17A and IL-6 decreased significantly in response to 3 and 1 mg/kg PCG (Figure 4B,C). MPL also reduced serum levels of IFN-γ, IL-17A, and IL-6 compared to vehicle control. PCG, at 3 mg/kg, reduced serum IL-6 levels more substantially than MPL (84.2% vs. 50.2%) (Figure 4C). Interestingly, serum levels of IL-10 were significantly augmented by 3, 1, and 0.3 mg/kg PCG compared to the vehicle control (133.1%, 113.3%, and 116.5%, respectively) (Figure 4D). In comparison with the vehicle control, 3, 1, and 0.3 mg/kg PCG also significantly increased serum concentrations of TGF-β1 (121.4%, 199.6%, and 161.0%, respectively). Notably, MPL did not significantly alter serum IL-10 and TGF-β1 levels (Figure 4D,E) 

### 2.9. Reduction of Spleen Size and Splenocytes by PCG

Based on the pivotal roles of PAR2 in regulating inflammation via multiple cellular processes [11], PCG was predicted to reduce the total splenic cells and rebalance the population of splenic CD4^+^ T cell subsets toward the low inflammatory status via PAR2 inhibition. As expected, both MPL and all doses of PCG clearly decreased spleen size compared to vehicle control (Figure 5A). Compared with the vehicle control, MPL and 3 and 1 mg/kg PCG significantly reduced spleen weights (52.4%, 40.9%, and 41.9%, respectively) (Figure 5B) and splenocyte counts (64.8%, 29.1%, and 30.0%, respectively) (Figure 5C).

### 2.10. Rebalance of Splenic CD4^+^ T Cell Subsets by PCG

MPL and 3 and 1 mg/kg PCG significantly reduced the populations of T_H_1 (56.6%, 40.3%, and 36.9%, respectively), T_H_17 (91.0%, 64.1%, and 50.7%, respectively), and T_H_2 cells (36.6%, 52.6%, and 46.0%, respectively) compared to the vehicle controls (Figure 5D,F). Meanwhile, 3 mg/kg PCG significantly enhanced the population of Treg cells compared to vehicle control by 23.6%. PCG at 1 mg/kg increased Treg cells by 16.4% compared to the vehicle control but the difference was not statistically significant. Additionally, 0.3 mg/kg PCG, which significantly increased both serum IL-10 and TGF-β1 levels, had no influence on Treg cell populations, and MPL did not affect Treg cell populations (Figure 5G). Furthermore, MPL and 3 and 1 mg/kg PCG significantly reduced the population of total splenic CD4^+^ T cells (Figure 5H). The populations of CD4^+^ T cell subsets in treatment groups are depicted in Appendix A.

### 2.11. PCG Reduces Anti-Double-Stranded DNA (dsDNA) and LN-Specific IgG Subclasses

Anti-dsDNA is the most critical and pathogenic autoantibody in nephritis in NZB/W F1 mice [26,27], and anti-dsDNA may reflect nephritis severity in NZB/W F1 mice [28,29]. MPL and 3 and 1 mg/kg PCG-treated mice exhibited significant reductions in the serum concentration of anti-dsDNA compared to that in vehicle-treated mice (67.5%, 52.1%, and 48.1%, respectively) (Figure 6A). Additionally, 3 mg/kg PCG significantly reduced serum concentrations of IgG1, IgG2b, and IgG3 subclasses compared to the vehicle control (32.0%, 33.8%, and 27.5%, respectively) (Figure 6B,D,E). Unlike PCG, MPL significantly decreased serum levels of IgG2a and IgG2b subclasses compared to vehicle control (53.8% and 42.8%) (Figure 6C,D). MPL and 3 and 1 mg/kg PCG-treated mice exhibited reduced kidney weights (Figure 6F). To investigate the effect of treatment on plasma cells, splenic CD19^+^CD138^+^B cells were counted. MPL, but not PCG, significantly decreased CD19^+^CD138^+^B cells (Figure 6G). 

### 2.12. PCG Reduces VCAM-1 and ICAM-1 Expression in Kidney Tissues of Lupus-Prone Mice

PCG strongly inhibited the PAR2-stimulated upregulation of VCAM-1 and ICAM-1 in the human podocyte cell line (Figure 1D). We further found that MPL and all doses of PCG significantly reduced the intensity of VCAM-1 and ICAM-1 immunohistochemical staining in kidney tissues (Figure 7A). MPL and 3, 1, and 0.3 mg/kg PCG diminished the expression of VCAM-1 (83.3%, 77.8%, 66.7%, and 66.7%, respectively) and ICAM-1 (83.3%, 77.8%, 72.2%, and 61.1%, respectively) in kidney tissues compared to levels in vehicle controls (Figure 7B). 

### 2.13. Toxicity of PCG 

The cytotoxicity of punicalin and PCG in NIH3T3 cells was evaluated after treatment for 24 h. Neither punicalin nor PCG affected cell viability at high concentrations (Appendix A). No significant changes in body weight were observed in mice treated with PCG (10 mg/kg, IP, every 48 h) for 10 days (Appendix A) [30]. No significant tissue damage was observed in the heart, liver, and lungs of mice in all groups at 30 weeks of age (Appendix A).

## 3. Discussion 

LN is the major cause of acute kidney injury and chronic kidney disease, leading to ESRD, which is related to all-cause mortality in SLE [2]. Even with aggressive induction and maintenance therapies, approximately 20% of patients with LN progress to ESRD [2,3]. Belimumab, a humanized monoclonal antibody that inhibits the binding of BAFF to its receptor, has recently been approved for the treatment of SLE [31,32]. However, its efficacy in severe LN is not well-supported by previous clinical trials [31,32,33,34,35,36]. Accordingly, a novel therapeutic agent is needed to induce and maintain remission.

LN may be initiated by the dysregulation of dendritic cells and autoantibody production by autoreactive B cells [2,3]. Additionally, an imbalance in CD4^+^ T cell subsets, such as increases in T_H_17 and follicular helper T (T_FH_) cells and a decrease in Treg cells, affects LN occurrence and relapse [37]. Increases in pro-inflammatory cytokines related to the pathogenesis of LN via the ERK/MAPK and NF-κB pathways may increase severity [18,38]. Therefore, newly developed immunosuppressive drugs should drive autoreactive immune cells and their intracellular signal transduction pathways towards an anti-inflammatory status. Based on this concept, our newly identified PAR2 antagonist, PCG, is likely to be an effective treatment modality for LN. PCG alleviated nephritis in lupus-prone mice by regulating serum pro-inflammatory and anti-inflammatory cytokines, rebalancing CD4+ T cell subsets, and diminishing LN-pathogenic autoantibodies.

A combination of high-dose glucocorticoids with either cyclophosphamide or mycophenolate mofetil is recommended as induction therapy for proliferative LNs, and a combination of glucocorticoids with either mycophenolate mofetil or azathioprine is also recommended as maintenance therapy [3,4,17]. Here, we applied MPL monotherapy (7 mg/kg/day) for induction and maintenance, instead of cyclophosphamide and mycophenolate mofetil, for two reasons. First, in real experimental settings using NZB/W F1 mice, an intravenous infusion of cyclophosphamide may induce cytotoxicity and systemic complications. Second, mycophenolate mofetil may be administered together with glucocorticoids owing to its lower toxicity than that of cyclophosphamide. In addition, glucocorticoid monotherapy and combination therapy with high-dose glucocorticoids and mycophenolate mofetil showed no significant difference in therapeutic efficacy [28,29].

Interestingly, PCG significantly increased serum IL-10 and TGF-β1 levels but MPL did not (Figure 4). IL-10 can be produced and secreted by IL-10-producing T_H_1 cells, T_H_2 cells, and T_H_17 cells along with IL-10-producing FoxP3^+^ or FoxP3^-^ Treg cells [39]. Thus, the elevated serum level of IL-10 might be due to an increase in IL-10-producing T_H_1 cells, T_H_2 cells, and T_H_17 cells or an increase in IL-10-producing Treg cells. Although we did not isolate and count helper T cells producing IL-10, PCG decreased the populations of splenic T_H_1, T_H_2, and T_H_17 cells and increased the population of splenic FoxP3^+^ Treg cells. Therefore, serum IL-10 might be largely secreted by Treg cells. Similar patterns in serum IL-10 and TGF-β1 support this assumption. PCG may alleviate nephritis in lupus-prone mice by rebalancing the population of splenic CD4^+^ T cells, particularly by increasing the population of Treg cells secreting both IL-10 and TGF-β1. Unlike other cytokines, the serum levels of IL-10 and TGF-β1 were not proportional to the PCG dose, suggesting that there are PAR2-dependent and-independent mechanisms underlying the PCG-mediated expression of IL-10 and TGF-β1. A previous study showed that PCG promotes IL-10 secretion by M2c-like macrophage polarization via the up-regulation of Heme oxygenase-1 (HO-1) [40]. In high-fat feeding conditions, *IL-10* mRNA levels in CD11b^+^ hepatic macrophages are significantly higher in PAR2^-/-^ mice than in wild-type mice [41].

Among various autoantibodies that can form immune complexes and are deposited in kidneys, anti-dsDNA is the most important for LN pathogenesis [26,27,28]. Thus, reducing the serum concentration of anti-dsDNA may prevent LN development and aggravation. Here, PCG significantly reduced the serum concentration of anti-dsDNA compared to that of the vehicle control (Figure 6A), suggesting that PCG improves nephritis in NZB/W F1 mice by directly inhibiting the production of the major LN-related autoantibody. Among IgG subclasses, IgG2a, IgG2b, and IgG3 are involved in LN pathogenesis and are associated with lupus-like diseases in NZB/W F1 mice [42,43,44]. PCG significantly decreased serum IgG2b and IgG3 along with IgG1 levels compared to levels in the vehicle group (Figure 6B,D,E). There was a discrepancy in the inhibition of IgG subclass production between MPL and PCG. Both IgG2a and IgG2b were significantly inhibited by MPL, whereas IgG2b and IgG3 were significantly suppressed by PCG. In addition, 3 mg/kg PCG significantly reduced serum IgG1 in NZB/W F1 mice, but IgG1 is not essential for the development of nephritis in NZB/W F1 mice [45]. Thus, PCG may minimize the extent of nephritis in NZB/W F1 mice by reducing serum levels of both anti-dsDNA antibodies and the pathogenic IgG subclasses, IgG2b and IgG3. 

PAR2 activation induces pro-inflammatory intracellular signaling via the ERK/MAPK and NF-κB signaling pathways [13,14,15]. In this study, we demonstrated that PCG significantly decreases the PAR2-induced activation of ERK1/2 and NF-κB, and thereby reduces the expression of VCAM-1 and ICAM-1 in vitro (Figure 1C,D). Furthermore, PCG significantly diminished VCAM-1 and ICAM-1 expression in vivo (Figure 7). These results are consistent with those of a previous study showing that ICAM-1 and VCAM-1 levels in colonic tissues are significantly attenuated in PAR2^-/-^ mice with TNBS-induced colitis compared to wild-type mice [46]. Therefore, PCG may improve nephritis in NZB/W F1 mice by the inhibition of PAR2-related inflammation, without causing adverse effects in major organs (Appendix A). 

Immunosuppressive drugs widely used to induce and maintain LN remission inhibit unspecific cell cycles or specific cells and signals [47]. However, PCG may directly affect autoreactive immune cells and intracellular signaling related to LN pathogenesis. PAR2 plays an important role in innate and adaptive immune responses, and its activation by serine proteases stimulates inflammatory-cytokine production and secretion in various cell types [48]. In LN, PAR2 can be activated by kidney-localized serine proteases and enhance pro-inflammatory cytokine production [12,49]. Theoretically, PCG may suppress circulating autoreactive DC, B, and T cells and end the vicious cycle of the recruiting and homing of immune cells to kidneys, thereby improving systemic complications in SLE, beyond LN. Thus, immune modulation via the inhibition of PAR2-mediated signaling by PCG may be beneficial for the treatment of LN. 

In summary, we provide evidence that PCG is a potent and selective antagonist of PAR2 and a potential therapeutic agent for LN. The inhibition of PAR2 by PCG reduced PAR2-induced ERK1/2 and NF-κB activation in the human podocyte cell line. Notably, PCG ameliorated kidney injury, proteinuria, and splenomegaly in NZB/W F1 mice and reduced ICAM-1 and VCAM-1 levels via PAR2 inhibition (Figure 8). Our results suggest that PCG can improve LN by inhibition of PAR2 in vitro and in vivo and is a potential therapeutic agent for LN.

## 4. Materials and Methods

### 4.1. Materials 

A library of 1279 natural products and punicalin was obtained from the Korea Chemical Bank (KRICT, Daejeon, Korea). PAR1-AP (TRLLR-NH_2_), thrombin, and trypsin were purchased from Tocris Bioscience (Bristol, UK). PAR2-AP (SLIGRL-NH_2_) was synthesized by Cosmo Genetech (Seoul, South Korea). Punicalagin (PCG), ellagic acid, and other chemicals were purchased from Sigma-Aldrich (St. Louis, MO, USA), unless otherwise indicated.

### 4.2. Cell Culture 

HaCaT and NIH3T3 cells were purchased from the Korean Cell Line Bank (Seoul, Korea) and were grown in DMEM (Hyclone, Logan, UT, USA) containing 10% FBS, 100 units/mL penicillin, and 100 μg/mL streptomycin at 37 °C and 5% CO_2_. Human podocyte cell line [50] was cultured in RPMI 1640 medium (Hyclone, Logan, UT, USA) with GlutaMax, 10% FBS, 100 units/mL penicillin, 100 μg/mL streptomycin, and 1% Insulin-Transferrin-Selenium (Life Technologies, Carlsbad, CA, USA) at 33 °C with 10% CO_2_ and differentiated at 37 °C with 10% CO_2_. Cells were grown in type I collagen-coated flasks.

### 4.3. Cell-Based Screening 

HaCaT cells were plated in 96-well microplates (Corning Inc., Corning, NY, USA) at 20,000 cells per well and cultured for 2 days. The cells were loaded using the Fluo-4 NW Calcium Assay Kit (Invitrogen, Carlsbad, CA, USA) according to the manufacturer’s protocol. Briefly, cells were incubated with 100 μL of assay buffer including Fluo-4 NW for 1 h; then, ~1279 natural compounds were applied to each well at 25 μM. After 10 min of incubation at 37 °C, the 96-well plates were transferred to a microplate reader (BMG Labtech, Ortenberg, Germany) equipped with a syringe pump and excitation/emission filters (485/520 nm). Fluo-4 fluorescence was measured for 2 s; then, 100 μL of phosphate-buffered saline (PBS) containing 5 nM trypsin was applied for 2 s to activate PAR2. Fluo-4 fluorescence changes due to alterations in the intracellular calcium concentration by PAR2 activation were recorded and analyzed using the MARS Data Analysis Software (BMG Labtech).

### 4.4. Intracellular Calcium Measurement

Intracellular calcium levels were measured in human podocytes using the Fluo-4 NW Calcium Assay Kit (Molecular Probes/Invitrogen) according to the manufacturer’s protocol. Human podocytes were treated with PCG, and intracellular calcium was induced by 10 μM PAR1-AP, 10 μM PAR2-AP, 5 nM trypsin (TR), and 100 Units/mL thrombin (Thr).

### 4.5. Immunoblotting

Human podocytes were washed twice with PBS and lysed using a cell lysis buffer (50 mM Tris-HCl (pH 7.4), 1% Nonidet P-40, 0.25% sodium deoxycholate, 150 mM NaCl, 1 mM EDTA, 1 mM Na_3_VO_4_, and protease inhibitor mixture) on ice. Whole-cell lysates were centrifuged at 13,000× *g* for 20 min at 4 °C, and protein extracts were separated on 4–12% Tris-glycine precast gels (Komabiotech, Seoul, Korea) and transferred to PVDF membranes (Millipore, Billerica, MA, USA). Membranes were blocked with 3% bovine serum albumin in TBST (Tris-buffered saline, 0.1% Tween 20) for 1 h and, then, incubated with anti-ERK1/2 (9101, 1:1000; Cell Signaling, Danvers, MA, USA), anti-p-ERK1/2 (9102, 1:1000; Cell Signaling), anti-P65 (sc-8008, 1:1000; Santa Cruz Biotechnology, Santa Cruz, CA, USA), anti-p-P65 (sc-136548, 1:1000; Santa Cruz Biotechnology), anti ICAM-1 (sc-8304, 1:1000; Santa Cruz Biotechnology), anti VCAM-1 (sc-8439, 1:1000; Santa Cruz Biotechnology), or beta-actin (sc-47778, 1:5000; Santa Cruz Biotechnology) antibodies. After washing the membranes three times with TBST, the membranes were incubated for 1 h with the appropriate HRP-conjugated secondary antibody at room temperature. Protein levels were detected with an ECL reagent using the FUSION SOLO imaging system (Vilber Lourmat, Marne-la-Vallée, France). ImageJ was used to quantify band intensity. 

### 4.6. Measurement of Cytokine Production In Vitro 

Human IL-6 ELISA Kit (ab178013), Interferon-gamma ELISA Kit (ab100538), Human TNF-alpha ELISA Kit (ab181421), and Human IL-8 ELISA Kit (ab108869) were purchased from Abcam (Cambridge, UK). Cytokine levels were determined according to the manufacturer’s protocol. Briefly, 50 µL of each sample and blank was placed in 96-well plates and incubated at room temperature for 1 h. Then, each well was washed 3 times with wash buffer and 50 μL of 1× biotinylated antibody for 30 min. TMB development solution was, then, added and incubated for 10 min. Then, 100 μL of stop solution was added, and absorbance was measured at 450 nm using an Infinite M200 Microplate Reader (Tecan Infinite M200 Pro; Tecan GmbH, Männedorf, Switzerland).

### 4.7. Lupus-Prone Mice and Treatment Protocol

NZB/W F1 mice exhibit first signs of nephritis at 13 weeks of age and fully developed nephritis at 22–24 weeks of age [26,27]. Female NZB/W F1 mice (21 weeks old) were purchased from Central Lab. Animal Inc. (Seoul, Korea) and housed in a specific pathogen-free barrier facility under standard sterile conditions. Mice were treated from 23 weeks of age and sacrificed at 30 weeks of age. PCG and methylprednisolone (Pfizer, Bruxelles, Belgium) were dissolved in PBS; mice in the vehicle group were injected with only PBS. Lupus-prone mice were assigned to vehicle (*n* = 8), 7 mg/kg/day MPL (*n* = 7), 0.3 mg/kg PCG (*n* = 5), 1 mg/kg PCG (*n* = 5), and 3 mg/kg PCG (*n* = 5) groups. A total weekly dose of MPL (49 mg/kg/week) was divided and intraperitoneally (IP) injected 3 times per week to avoid infectious peritonitis. PCG was also IP injected 3 times per week. PBS was IP injected to vehicle-treated mice according to the same schedule.

### 4.8. Measurement of Proteinuria 

Proteinuria was measured twice per week using albumin reagent strips for spot urine (Yongdong Pharmaceutical Co., Yongin, Korea). The amount of proteinuria was determined by the following semi-quantitative scoring system: 0 = none or trace; 1+, ≤00 mg/dL; 2+, ≤300 mg/dL; 3+, ≤1000 mg/dL; 4+, ≥1000 mg/dL [28].

### 4.9. Measurement of Serum Creatinine

Serum creatinine concentration was measured using the QuantiChrom™ Creatinine Assay Kit (DICT-500; BioAssay Systems, Hayward, CA, USA) through the optimized Jaffe method with colorimetric creatinine determination at 510 nm.

### 4.10. Renal Histopathology

Mice were perfused with 10 mL of PBS through the left ventricle through an incision. Kidneys were isolated from all mice after sacrifice. Formalin-fixed kidney specimens were embedded in paraffin, cut into 4-μm-thick sections, and stained with periodic acid–Schiff (PAS) according to conventional procedures. Glomerular, tubular, and vascular damage levels were scored semi-quantitatively on a four-point scale by two independent pathologists, as described in our previous report [28,29].

### 4.11. Immunofluorescence

Kidneys were embedded in OCT compound and frozen at −70 °C. Samples were cut into 5-μm-thick sections, fixed in 4% paraformaldehyde, and washed 3 times in cold PBS. Nonspecific binding was blocked with 1% normal goat serum in PBS with Tween 20 for 30 min. Sections were incubated with goat anti-mouse IgG (1:100; Serotec, Oxford, UK) and rabbit anti-mouse complement 3 (C3) (1:100; Abcam) at 4 °C overnight, washed 3 times with PBS, incubated with Alexa Fluor 568–labeled donkey anti-goat IgG and Alexa Fluor 488-labelled donkey anti-rabbit IgG (each 1:100; Invitrogen) at room temperature for 1 h, and washed 3 times with PBS. Sections were mounted in a mounting solution (Vector Laboratories, Burlingame, CA, USA) and examined under a laser scanning confocal microscope (LSM 780; Carl Zeiss, Oberkochen, Germany). Two pathologists independently assigned semi-quantitative scores indicating the intensity and distribution of immunofluorescence staining for IgG and C3, on a scale ranging 0–3, in which 0 = no staining, 1 = weak staining, 2 = moderate staining, and 3 = strong staining. The average score obtained by the 2 pathologists was determined.

### 4.12. Serum Cytokine Measurement 

Serum samples were stored at −80 °C. Serum cytokines were measured by enzyme-linked immunosorbent assay (ELISA). IFN-γ, IL-17A, IL-6, and IL-10 levels were quantified using a ProcartaPlex Multiplex Immunoassay Kit (Invitrogen) according to the manufacturer’s instructions. Serum TGF-β1 was quantified using a DuoSet ELISA Kit (DY1679; R&D Systems, Minneapolis, MN, USA).

### 4.13. Serum Anti-dsDNA and Immunoglobulin G (IgG) Subclass Measurement 

The anti-dsDNA IgG concentration was measured by sandwich ELISA (Alpha Diagnostic International, San Antonio, TX, USA). IgG subclasses were analyzed using a ProcartaPlex Multiplex Immunoassays Mouse Immunoglobulin Isotyping Kit (Invitrogen). 

### 4.14. Immunohistochemistry 

Immunohistochemical analysis was performed using a Vectastain ABC Kit (Vector). Tissues were stained with goat antibodies against mouse intercellular adhesion molecule 1 (ICAM-1) (clone YN1/1.7.4; 1:100 dilution, Abcam) and vascular cell adhesion molecule 1 (VCAM-1) (clone M/K-2; 1:100 dilution, Abcam) overnight at 4 °C and, then, with biotinylated secondary antibodies linked to streptavidin–peroxidase complex (SantaCruz Biotechnology) for 1 h. Sections were developed using amino-ethylcarbazole (Dako, Carpinteria, CA, USA) and counterstained with hematoxylin. Images were captured under a light microscope (Olympus, Tokyo, Japan). A semiquantitative score indicating the intensity and distribution of ICAM-1 or VCAM-1 staining on glomeruli and tubules was assigned as follows: 0 normal or slight staining; 1 = focal, mildly increased staining; 2 = focal, moderately increased staining; and 3 = diffuse, markedly increased staining. Scores were determined independently by 2 pathologists in a blinded manner, and the average score was calculated.

### 4.15. Statistical Analyses 

All values and error bars were calculated from duplicate estimates and are presented as means ± SEM of two independent experiments. Statistical analyses of differences among groups were performed using Student’s *t*-test, with the exception of survival data. A Kaplan–Meier survival analysis was performed using the log-rank test.

## Figures and Tables

**Figure 1 ijms-21-04975-f001:**
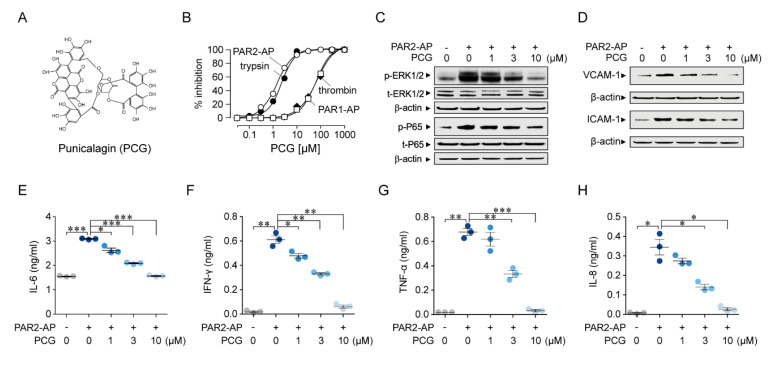
Characterization of punicalagin, a novel PAR2 antagonist, in the human podocyte cell line. (**A**) Chemical structure of punicalagin (PCG). (**B**) Summary of dose responses (mean ± SEM, *n* = 5–6). (**C**) Western blot analysis of phospho-ERK1/2 (p-ERK1/2), total ERK1/2 (t-ERK1/2), phospho-P65 (p-65) and total P65 (t-p65). The indicated concentrations of PCG were applied 30 min prior to PAR2 activation by PAR2-AP. (**D**) Inhibition of the PAR2-induced upregulation of VCAM-1 and ICAM-1 by PCG. VCAM-1 and ICAM-1 expression levels were detected at 6 h after PAR2 activation. (**E**–**H**) The indicated concentrations of PCG were applied 30 min prior to the application of PAR2-AP. IL-6, IFN-γ, TNF-α, and IL-8 concentrations were measured at 12 h after PAR2 activation (mean ± SEM, *n* = 3). * *p*  <  0.05, ** *p*  <  0.01, and *** *p*  <  0.001. Two-tailed Student’s *t*-test (**E**,**F**).

**Figure 2 ijms-21-04975-f002:**
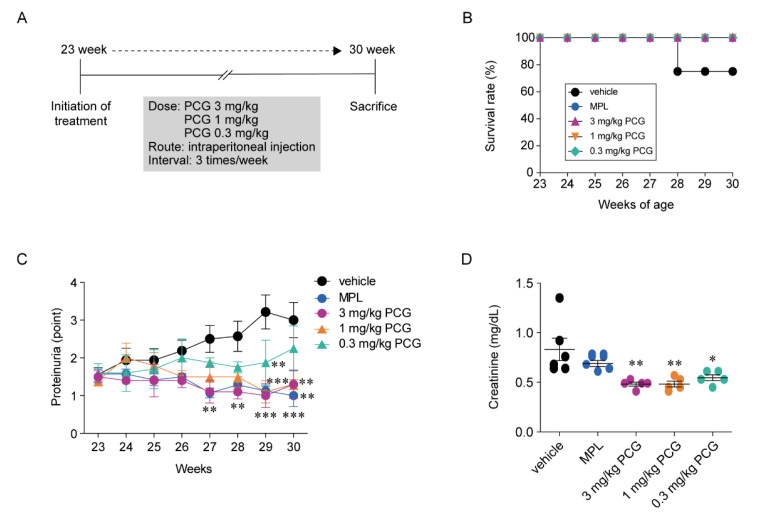
PCG preserves renal function in lupus-prone mice. (**A**) Treatment schedule for lupus-prone mice with phosphate-buffered saline (vehicle), methylprednisolone (MPL) (7 mg/kg/day), and 0.3 mg/kg, 1 mg/kg, and 3 mg/kg PCG is presented. PCG was applied 3 times a week. (**B**) The survival rate of lupus-prone mice treated with vehicle, MPL, and three doses of PCG (mean ± SEM, *n* = 5–8/group). (**C**) The amount of proteinuria in lupus-prone mice was measured twice a week using albumin reagent strips (mean ± SEM, *n* = 5–8/group). (**D**) Serum creatinine concentration was measured through a kinetic colorimetric method (mean ± SEM, *n* = 5–7/group). * *p* < 0.05, ** *p* < 0.01 and *** *p* < 0.001 versus vehicle. Two-tailed Student’s *t*-test (**C** and **D**).

**Figure 3 ijms-21-04975-f003:**
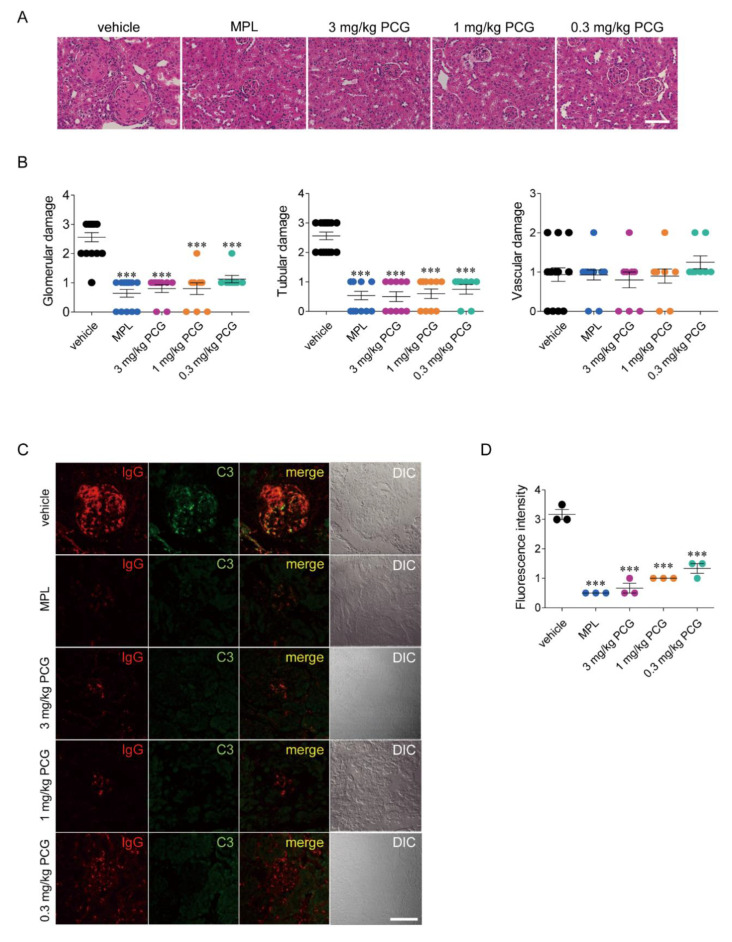
PCG improves histological damage and reduces immune-complex deposition in lupus-prone mice. (**A**) Periodic acid–Schiff (PAS) staining of kidney tissues of NZB/W F1 mice at 30 weeks of age. Mice were treated with MPL and PCG 3 times a week. Bar = 50 μm. (**B**) Summary of glomerular, tubular, and vascular damage in kidney tissues (mean ± SEM, *n* = 5–7/group). (**C**) The deposition of immune-complex was evaluated by immunofluorescence staining of IgG (red) and complement 3 (green). Bar = 50 μm. (**D**) The intensity of fluorescence was scored on a 4-point scale (mean ± SEM, *n* = 5–7/group). *** *p* < 0.001 versus vehicle. Two-tailed Student’s *t*-test (**B** and **D**).

**Figure 4 ijms-21-04975-f004:**
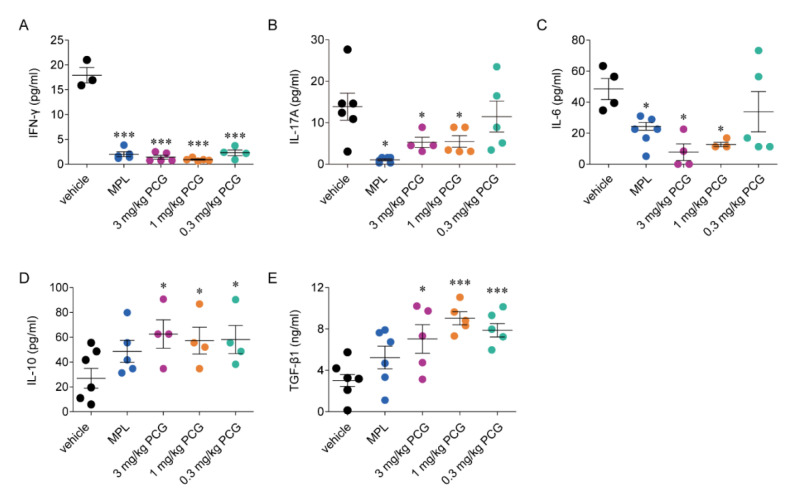
PCG decreases pro-inflammatory cytokines and increases anti-inflammatory cytokine levels in NZB/W F1 mice. (**A**–**E**) Plasma concentrations of IFN-γ (**A**), IL-17A (**B**), IL-6 (**C**), IL-10 (**D**), and TGF-β1 (**E**) were measured in NZB/W F1 mice at 30 weeks of age (mean ± SEM, *n* = 3–7/group). Mice were treated with MPL and PCG 3 times a week. * *p* < 0.05 and *** *p* < 0.001 versus vehicle. Two-tailed Student’s *t*-test (**A**–**E**).

**Figure 5 ijms-21-04975-f005:**
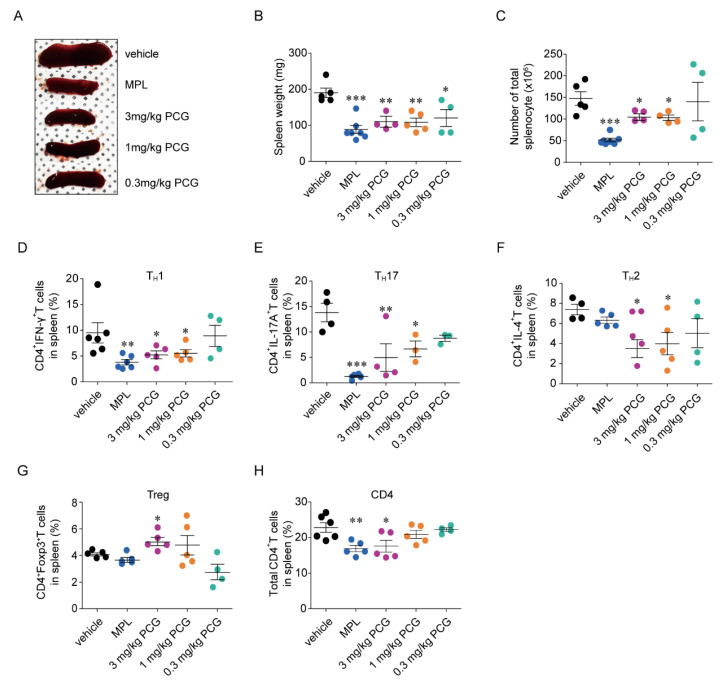
PCG induces a functional rebalance of splenic T cell subsets in NZB/W F1 mice. (**A**) Representative images of the NZB/W F1 mouse spleen at 30 weeks of age. (**B**) Summary of spleen weight (mean ± SEM, *n* = 4–7/group). (**C**) Total splenocyte counts (mean ± SEM, *n* = 4–7/group). (**D**–**H**) Splenocytes were prepared from the spleens of NZB/W F1 mice and activated with anti-CD3/CD28 for 72 h. The numbers of CD4^+^ T cells expressing IFN-γ (**D**), IL-17A (**E**), IL-4 (**F**), FoxP3 (**G**), and total CD4^+^ T cells (**H**) were analyzed (mean ± SEM, *n* = 3–7/group). * *p* < 0.05, ** *p* < 0.01 and *** *p* < 0.001 versus vehicle. Two-tailed Student’s *t*-test (**B**–**H**).

**Figure 6 ijms-21-04975-f006:**
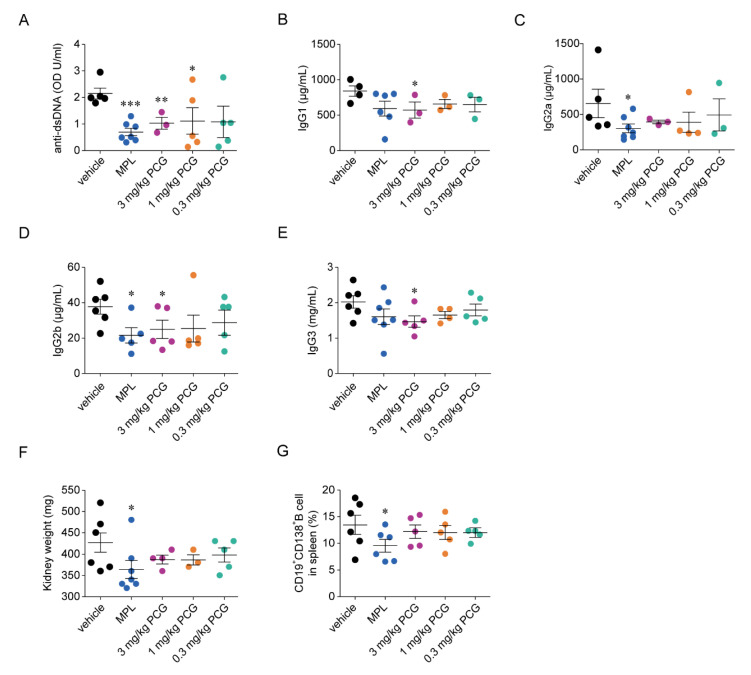
PCG reduces the levels of anti-dsDNA, pathogenic IgG subclasses, and plasma cells. (**A**) Serum concentration of total anti-dsDNA IgG was measured in NZB/W F1 mice at 30 weeks of age (mean ± SEM, *n* = 3–7/group). (**B**–**E**) IgG subclasses were analyzed (mean ± SEM, *n* = 3–7/group). (**F**) Summary of kidney weights (mean ± SEM, *n* = 3–7/group). (**G**) Mouse splenocytes were isolated, and CD19^+^CD138^+^ plasma cells in spleens were detected (mean ± SEM, *n* = 5–7/group). * *p* < 0.05, ** *p* < 0.01 and *** *p* < 0.001 versus vehicle. Two-tailed Student’s *t*-test (**A**–**G**).

**Figure 7 ijms-21-04975-f007:**
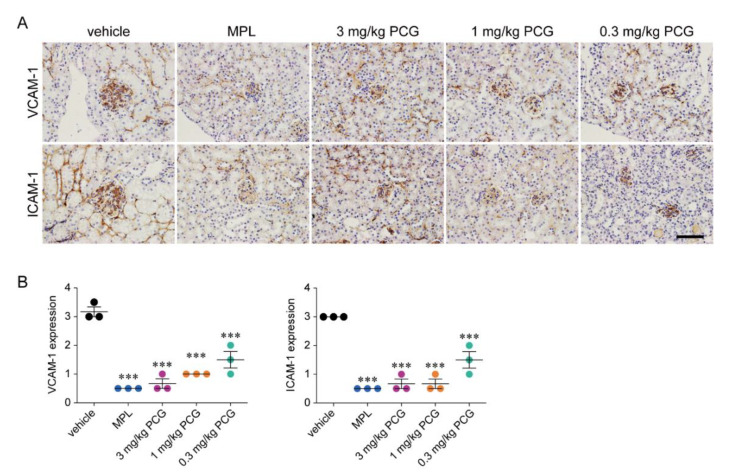
PCG suppresses the expression of VCAM-1 and ICAM-1 in kidney tissues of lupus prone mice. (**A**) Immunohistochemical staining of VCAM-1 and ICAM-1 in kidney tissues in NZB/W F1 mice at 30 weeks of age. Scale bar = 50 μm. (**B**) The intensity of VCAM-1 and ICAM-1 staining on glomeruli and tubules was assigned based on a 4-point scale (mean ± SEM, *n* = 4–8/group). *** *p* < 0.001 versus vehicle. Two-tailed Student’s *t*-test (**B**).

**Figure 8 ijms-21-04975-f008:**
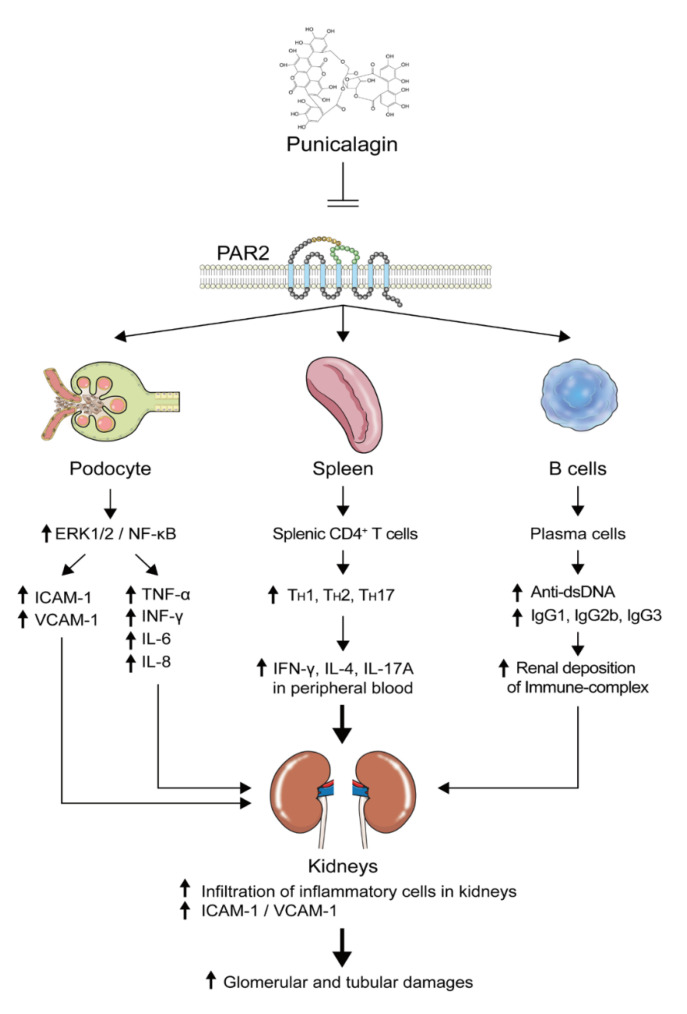
Schematic overview of the inhibitory effect of PCG on lupus nephritis in human podocytes and NZB/W F1 mice.

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
