# Peer review of "Punicalagin Ameliorates Lupus Nephritis via Inhibition of PAR2"

_ijms, 2020, doi:10.3390/ijms21144975_

Round 1

Reviewer 1 Report

-The hypothesis is correct

-The experiments are well planned except that some control subjets seem to have developed the desease with less intensity, a mouse strain control group would have elucidated that unknown (a negative control). 

-No loading controls for ERK, pERK, P65, pP65 

wbs was shown.

-Photographs should be shown in the same power

-Many subjets are missing in the final result, this decrease its quality and relevance regardless of the statistical significance 

Author Response

We greatly appreciate the editor’s and reviewers’ efforts to carefully review our manuscript and the valuable comments and suggestions offered for the improvement of the manuscript. We have made each of the suggested revisions. The points of criticism raised by the reviewers were addressed by a point-by-point response. Changes in the manuscript text are highlighted in red color font. We hope the editor and reviewers will agree with our points, and find our revised manuscript suitable for publication.

1. The experiments are well planned except that some control subjects seem to have developed the disease with less intensity, a mouse strain control group would have elucidated that unknown (a negative control).

Response. Thank you for the comment. In our previous study, we showed that NZB/NZW F1 mice can spontaneously develop lupus-like disease at 23 weeks old mice, and saline- or PBS-treated control group had a disease severity similar to that of this experiment (reference 28, 29). In the spontaneously developing mouse model of lupus nephritis, the disease severity varies from mouse to mouse.

2. No loading controls for ERK, pERK, P65, pP65 WBs were shown.

Response. Sorry for the confusion. In Figure 1C, ERK1/2 and P65 were total ERK1/2 and total P65. In Figure 1C, the ERK1/2 and P65 were replaced with t-ERK1/2 and t-P65, respectively, and b-actin blots were added.

3. Photographs should be shown in the same power

Response. Thank you for the comment. We carefully revised all the photographs in the revised manuscript.

4. Many subjects are missing in the final result, this decrease its quality and relevance regardless of the statistical significance.

Response. Thank you for the comment. NZB/NZW F1 mice spontaneously develop lupus-like disease at ~23 weeks and the disease severity varies from mouse to mouse, and some of the NZB/NZW F1 mice died from renal failure during the experiment. Thus some subjects are missing in the final result. The mortality rates for NZB/NZW F1 mice were similar to those in our previous experiments (reference 28, 29).

Reviewer 2 Report

The manuscript describes an elegant and well presented study showing the potentialities of using a novel molecule to treat lupus nephritis. Exps have been conducted in both in vitro and in vivo models  and this is appreaciated.  

I have only a suggestion:

1) The study showing renal protection are pretty convincing. Punicalagin is a potent anti-inflammatory agent. I only would suggest to provide further evidence indicating that punicalagin-dependent kidney protection is mediated by PAR2 antagonism. WOuld you consider to test anti-PAR2 antibody or other known antagonists?

Author Response

We greatly appreciate the editor’s and reviewers’ efforts to carefully review our manuscript and the valuable comments and suggestions offered for the improvement of the manuscript. We have made each of the suggested revisions. The points of criticism raised by the reviewers were addressed by a point-by-point response. Changes in the manuscript text are highlighted in red color font. We hope the editor and reviewers will agree with our points, and find our revised manuscript suitable for publication.

1. The study showing renal protection are pretty convincing. Punicalagin is a potent anti-inflammatory agent. I only would suggest to provide further evidence indicating that punicalagin-dependent kidney protection is mediated by PAR2 antagonism. Would you consider to test anti-PAR2 antibody or other known antagonists?

Response. Thank you for the comment. As reviewer’s opinion, PAR2 antagonism by using anti-RAR2 antibody or other PAR2 antagonists may provide more useful information. In the present study, we investigate the effect of punicalagin (a novel PAR2 antagonist) on lupus nephritis because of the limited resource, and previous studies were shown that PAR2 antagonism effectively reduced inflammatory response in various inflammatory disease models including glomerulonephritis. In this study, we did not provide the direct evidence indicating that punicalagin-dependent kidney protection is mediated by PAR2 antagonism. However, the authors found out that punicalagin is a novel potent PAR2 antagonist, and it strongly decreased PAR2-mediated ERK1/2 and NF-kB signaling and expression of ICAM-1 and VCAM-1 in human podocyte cells. In addition, previous studies showed that PAR2 can be activated by kidney-localized serine proteases in renal injury and enhance pro-inflammatory cytokine production (reference 12). Thus the authors believe that punicalagin can ameliorates lupus nephritis, at least in part through PAR2 inhibition.

Round 2

Reviewer 1 Report

Nothing to add to the previous review and all questions have been answered